# Design of Fiber-Tip Refractive Index Sensor Based on Resonant Waveguide Grating with Enhanced Peak Intensity

**Yicun Yao** [1,2,*] **, Yanru Xie** [1,2] **, Nan-Kuang Chen** [1,2] **, Ivonne Pfalzgraf** [3] **, Sergiy Suntsov** [3] **, Detlef Kip** [3] **and Yingying Ren** [4]

1. School of Physical Science and Information Technology, Liaocheng University, Liaocheng 252059, China; xieyanru@lcu.edu.cn (Y.X.); nankuang@gmail.com (N.-K.C.)
2. Key Laboratory of Optical Communication Science and Technology of Shandong Province, Liaocheng University, Liaocheng 252059, China
3. Faculty of Electrical Engineering, Helmut Schmidt University, 22043 Hamburg, Germany; pfalzgri@hsu-hh.de (I.P.); suntsov@hsu-hh.de (S.S.); kip@hsu-hh.de (D.K.)
4. Shandong Provincial Engineering and Technical Center of Light Manipulations & Shandong Provincial Key Laboratory of Optics and Photonic Device, School of Physics and Electronics, Shandong Normal University, Jinan 250014, China; ryywly@sdnu.edu.cn
* Correspondence: yaoyicun@lcu.edu.cn

**Featured Application: Probe-like, multi-channel fiber biochemical sensor with ultra-small footprint.**

**Abstract:** Resonant waveguide gratings (RWG) are widely used as on-chip refractometers due to their relatively high sensitivity to ambient refractive index changes, their possibility of parallel high-throughput detection and their easy fabrication. In the last two decades, efforts have been made to integrate RWG sensors onto fiber facets, although practical application is still hindered by the limited resonant peak intensity caused by the low coupling efficiency between the reflected beam and the fiber mode. In this work, we propose a new compact RWG fiber-optic sensor with an additional Fabry-Pérot cavity, which is directly integrated onto the tip of a single-mode fiber. By introducing such a resonant structure, a strongly enhanced peak reflectance and improved figure of merit are achieved, while, at the same time, the grating size can be greatly reduced, thus allowing for spatial multiplexing of many sensors on a tip of a single multi-core fiber. This paves the way for the development of probe-like reflective fiber-tip RWG sensors, which are of great interest for multi-channel biochemical sensing and for real-time medical diagnostics.

**Keywords:** resonant waveguide grating; fiber sensor; F-P cavity

## 1. Introduction

Fiber sensors are characterized by their compact size, remote sensing ability and immunity to electromagnetic interference [1]. Currently, most fiber refractometers operate based on evanescent field interaction with a medium surrounding the fiber [2]. Usually, for such sensors, the dynamic range for refractive index detection is naturally limited by the refractive index of silica fiber to satisfy the total internal reflection condition. This gives rise to another issue of exciting and collecting evanescent light, or cladding modes, when higher index fluid samples are used. Moreover, it is also difficult to achieve multi-channel detection in practical applications with composite fluids like blood. In contrast, fiber-integrated sensors can be good candidates to realize both a wide dynamic index sensing range and multi-channel detection ability. Tiny open Fabry-Pérot (FP) cavities, fabricated by, e.g., the splicing of fiber segments or precision dicing close to or directly at the fiber tip, have attracted a great deal of attention due to their very high sensitivities [3,4]. However, their multiplexing abilities are typically limited to a small number of sensing elements located at different spatial positions along the fiber [5]. To date, this has hindered their use in biomedical applications where small amounts of liquid samples need to be

tested. The proposed RWG sensor on a fiber tip represents an interesting alternative to other types of fiber sensors. Although typical sensitivities are lower than that of, e.g., the FP cavities' sensor, this concept provides the principal advantage of multiplexing many sensors at a single fiber tip for testing the same small volume of a sample liquid.

RWGs, sometimes named guided mode resonant gratings (GMRG) or leaky mode resonant gratings (LMRG), are structures composed of higher refractive index waveguiding layers with a grating on the surface of lower index substrates. The study of this resonant grating phenomenon can be traced back to 1941, when Fano suggested that some of Wood's anomaly may be attributed to the excitation of surface waves [6]. This was further theoretically demonstrated by Hessel and Oliner, in 1965, by applying a metallic grating model [7]. Later in 1973, Neviere et al. studied these anomalies in dielectric gratings and related them to the excitation of leaky waveguide modes [8]. When a wave is incident on an RWG, the first-order diffracted light can be coupled into a leaky waveguide mode for a certain incident angle and wavelength. Consequently, leaky waves interfere destructively with the 0th-order transmitted wave to result in sharp resonant peaks in the reflection spectrum, which can reach up to 100% modulation [9]. By selecting proper waveguide and grating parameters, both the peak position and its width can be adjusted. Benefiting from its unique dispersion characteristics and resonant properties, various RWG applications have been developed, including narrowband and broadband wavelength filters, polarizers, color mixers, and devices for the enhancement of nonlinear processes [10–14].

RWGs can also be applied as refractive index sensors because the resonant wavelength is sensitive to index changes in an ambient environment [15–18]. The sensitivity of RWG sensors is usually in the range from several tens to several hundreds of nanometers per refractive index unit (nm/RIU), which is sufficient for most bio-sensing applications [19]. In addition, RWG sensors offer a nearly linear response for relatively large refractive index ranges (depending on waveguide and grating materials) and suffer less from cross-sensitivity due to temperature fluctuations [20]. Consequently, RWG sensors are widely studied for on-chip devices and high-throughput detection systems that are available for living cell sensing [21].

The proposed fiber refractometers can be realized by depositing a RWG directly on the fiber facet, offering a much larger index detection range compared to lateral detection fiber sensors. Furthermore, multi-channel detection can be achieved using multi-core fibers. However, integrating RWG sensors on a fiber-tip is still challenging. One typical difficulty is due to the small size of the fiber core. In the case of on-chip RWGs, relatively long gratings are needed to achieve strong enough resonance. Conversely, with a fiber-tip RWG, only power from a limited number of grating periods around the core can be coupled back to the core mode resulting in a weak peak intensity, which may be difficult to detect considering inherent background noise. In their work from 2000 [22], Wawro and Magnusson demonstrated a ~18% transmittance notch obtained with silicon nitride ($Si_3N_4$) grating films and deposited on a fiber tip using interference lithography, although the read-out of the signal was difficult because of the high level of background noise. Later, in 2018, regarding enhanced peak intensity, the same group reported the realization of an improved fiber-tip RWG sensor based on a multi-mode fiber (MMF) with the collection of the transmitted signal [23].

In practical sensing applications, probe-type fiber RWGs, with the use of the reflected broadband signal for detection, are preferable both in biochemical sensing and for real-time medical diagnostics. Since low coupling efficiency between single-mode fiber (SMF) and MMF would hinder direct employment of existing fiber-optic interrogation systems for input and read-out of signals, it is of importance to further enhance the resonant properties of RWG filters on an SMF facet.

In this work, we propose the use of an FP cavity surrounding the grating to improve the reflection peak intensity of the fiber-tip RWG sensors. At the same time, the footprint of the RWG sensor is strongly reduced. Using resonant enhancement, the designed novel RWG sensor has a reflection peak intensity of about three times that of a normal quasi-

infinite RWG. The high sensitivity, along with the compact sensor's dimensions, would allow for multi-channel sensing using multi-core optical fibers and could have good potential for refractive index sensing in small volumes such as in biomedical applications.

## 2. Structure and Working Principle

A resonance occurs with RWGS, when the phase matching condition is satisfied along the direction that is perpendicular to the gratings' lines, between the first diffraction order and the leaky waveguide mode, given by the following formula:

$$n \sin \theta + m \frac{\lambda}{\Lambda} = n_{eff}$$

where $n$ is the refractive index of the incident medium, $\theta$ the angle of incidence, $m$ the diffraction order, $\Lambda$ the grating period, $\lambda$ the vacuum wavelength, and $n_{eff}$ the real part of the effective refractive index of the waveguide mode. For the calculation of $n_{eff}$, effective medium theory (EMT) can be applied to substitute a grating with a homogeneous thin film [24]. In the case of the fiber-tip RWG in this work, $\theta$ is regarded as zero.

A simplified two-dimensional (2D) infinite grating and a plane wave incident situated vertically onto the grating were employed as a starting point for the sensor's design. A $Si_3N_4$ with a refractive index of ~2.0 at around 1550 nm wavelength was chosen for the waveguide and grating, due to good adherence between the quartz and $Si_3N_4$ layers [25]. The substrate material index was set to 1.452, the same as that of the fiber core. In aqueous solution detection, the surrounding refractive index was set to 1.33. Numerical tools (DiffractMOD module, RSoft 2021) based on the rigorous coupled mode analysis method (RCWA) were employed for determination of the exact structural parameters. The detailed design parameters are presented in Table 1, where $d_1$ and $d_2$ are the thickness of the waveguide and grating layers, respectively, and $c$ is the duty circle. Polarization was set to TE with the electric field component $E_y$ established parallel to the grating grooves.

**Table 1.** Sensor's design parameters used for the simulations.

| Parameter | $d_1$ | $d_2$ | $\Lambda$ | $c$ | $n_{core}$ | $n_{clad}$ | $n_{env}$ | $n_{wg}$ |
|---|---|---|---|---|---|---|---|---|
| Value | 0.18 μm | 0.2 μm | 0.95 μm | 0.7 | 1.452 | 1.447 | 1.33 | 2.0 |

An example of RCWA simulation results is shown in Figure 1. In relation to the given parameters, the peak position of the resonance was found to be at 1543.8 nm wavelength, as seen in Figure 1a. A typical standing wave pattern, due to the second-order Bragg reflection, is observed at the resonant wavelength and is shown in Figure 1b.

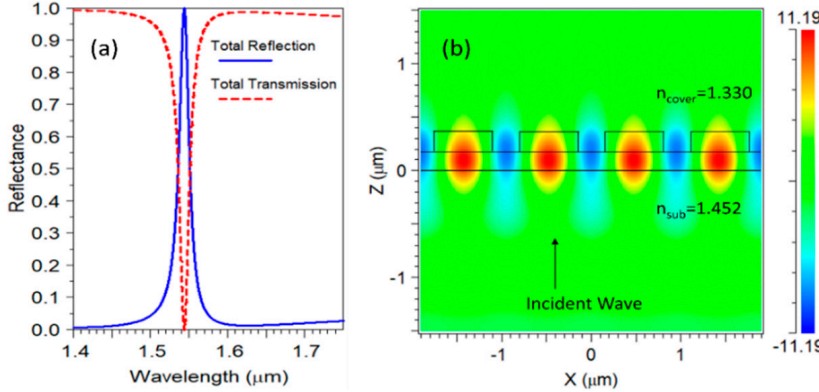

**Figure 1.** RCWA results for the 2D RWG. (**a**) Reflection (solid blue) and transmission (dashed red) spectra; (**b**) Electric field component Ey at the resonant wavelength of 1543.8 nm (normalized to the incident field).

The designed fiber-tip RWG sensor is presented schematically in Figure 2a,b. A Corning SMF28e fiber was chosen with a core diameter of 8.2 μm, and core and cladding indices of 1.452 and 1.447, respectively. The practical implementation of this sensor can be achieved using, e.g., photolithography with subsequent etching, or direct focused ion beam (FIB) structuring. For both methods, the waveguiding high-index $Si_3N_4$ layer needs to be deposited first using, e.g., sputtering or e-beam evaporation. Using a high-resolution photoresist (e.g., AZ@ MIR 701) and an interference lithography Lloyd's mirror setup with a 400 nm laser, quasi-infinite photoresist grating can be implemented over the whole area of the 125 μm diameter fiber cladding and subsequently transferred into the $Si_3N_4$ layer using dry etching. However, fabrication of the gold (Au) metal mirrors within the required positioning accuracy would be quite challenging (see Figure 2a). In contrast, FIB milling allows for high-resolution and high-positioning-accuracy fabrication of an RWG grating with a limited number of bars and metal mirrors on its sides. The fabrication steps using this method would consist of: low-ion-current FIB milling of a grating in the $Si_3N_4$ layer over the fiber core region; milling two trenches with a depth h and their inner walls in the middle of the side grating bars, as shown in Figure 2a,b; ion-beam assisted metal layer deposition on the inner trenches' sidewalls to form an FP resonator using appropriate sample pre-tilt with respect to the ion beam. To achieve good reflectance for the mirrors, Au layer thicknesses of 150 nm would be sufficient. The designed RWG sensor has a small footprint of about $15 \times 17$ μm$^2$, such that multiplexing of many sensors on a facet of a multi-core fiber could be achieved. An example of a multiplexed RWG sensor with 19 parallel sensors integrated on the facet of a commercial 19-core fiber is shown schematically in Figure 2c.

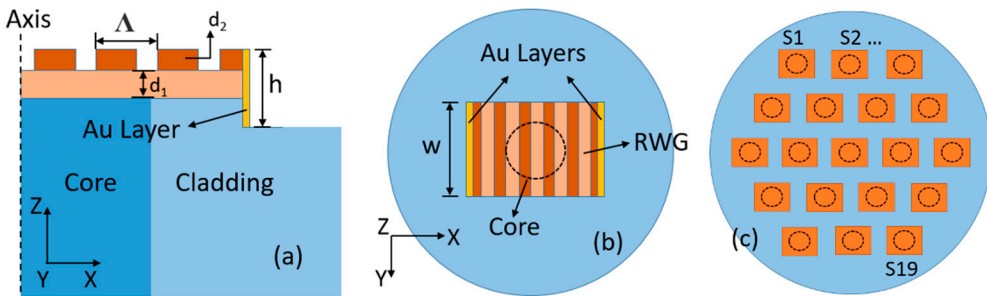

**Figure 2.** Schematic of the designed fiber-tip RWG sensor: (**a**) cross-section perpendicular to the grating lines, (**b**) top view, and (**c**) multiplexed sensor array on the facet of a typical 19-core fiber. (**a**) The black dashed line represents the central (core) axis of the fiber, and only half of the sensor's geometry is presented for simplicity.

The operation principle of the designed sensor is as follows. When a guided waveguide mode impinges upon the mirror, it will be reflected, and the leaked energy can be coupled back to the core fiber mode with high efficiency. The use of mirrors is equal to the application of periodic boundary conditions in simulations for waves propagating inside the FP cavity. This kind of restriction is much stronger compared to the bare Bragg reflection offered by a uniform grating, thus helping to achieve strong resonant peak intensities with a limited number of grating periods.

## 3. Results and Discussion

Due to the relatively large grating layer thickness, far from being treated within perturbation approximation, a numerical modeling approach is needed to obtain precise results. Here, the performance of the designed fiber-tip RWG is first studied with a 2D finite-difference time domain (FDTD) method using commercial software (FullWave, RSoft 2021), and the design parameters are listed in Table 1. The Au mirrors are located at the ninth grating bar on both sides, with the center of the fiber core in the middle of the central grating notch. In the simulations, the TE-polarized core fiber mode (electric field

along grating grooves), located 0.6 µm below the fiber/waveguide interface, was used as the incident beam. Two power monitors in the xy-plane were located 1.15 µm below the fiber/waveguide interface to collect the reflected power, one for the total reflected power and another one for the mode reflection, with the latter defined by the overlapping integral between the fiber core mode and the field at that specific xy-plane. The mesh sizes were set to be 20 nm for both x- and z-direction. Within the waveguide, for the grating and Au layers, as well as for areas around all structural edges, non-uniform meshes were applied with finer grid sizes assuring results with good convergence. Perfectly matched layer (PML) boundaries were employed for ±z and +x, while symmetric boundaries were employed for –x, to save computational resources.

The calculated reflection spectrum is shown in Figure 3a. The main resonant peak at ~1529.5 nm is blue-shifted compared to the RCWA result (green curve). This blue shift may be attributed to both the finite beam size of the core mode and the grating length, as discussed in Reference [26]. With the use of Au layers to create a FP cavity, the achieved resonant mode reflectance was ~74.8%, which is quite close to the total reflectance of ~79.4% (see Figure 3a). The mode reflectance is specified as the part of the total incident energy coupled back to the core mode of the fiber, while the total reflectance is the part of the total incident energy diffracted back by the grating. Apparently, most of the reflected power can be coupled back to the core mode. It is worth noticing that, besides the main resonant peak, a secondary peak is observed at ~1634.0 nm for both the mode and the total reflectance. This side peak cannot be found in the RCWA curve for the uniform grating, and this could be attributed to the FP resonance behavior.

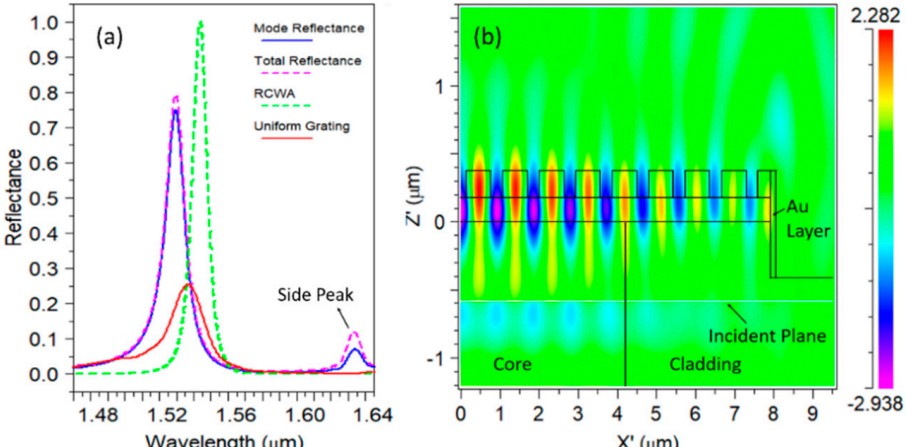

**Figure 3.** (**a**) Reflection spectrum for the designed fiber-tip RWG, with mode reflectance (energy coupled into the backward propagating fiber mode, solid blue curve), total reflectance (total energy diffracted back by the grating, dashed magenta curve), reflectance by 2D-RCWA (dashed green curve). For comparison, mode reflectance for a uniform grating without FP cavity is also shown (red); (**b**) Profile of $E_y$ electric field component for the peak wavelength of 1529.5 nm.

The mode reflectance for a uniform grating on the fiber facet with the same design parameters is presented in Figure 3a by a red curve. In the corresponding simulation, the +x domain, expanded as grating bars far away from the core area, may also contribute to mode reflectance because the grating itself also serves as a second-order Bragg grating, and coupling occurs into waveguide modes propagating along both positive and negative x-directions. The mode reflectance for the uniform grating was found to be ~25.4%. In comparison, the use of a FP cavity improved the peak intensity approximately three times, and, at the same time, the peak FWHM (full width at half maximum) was reduced from ~23.0 nm to 13.7 nm due to the improved Q factor of the FP cavity. Such an increase in peak intensity in conjunction with a decreased FWHM is crucial for signal identification in sensing applications. The electric field profile at the resonant wavelength of 1529.5 nm is presented in Figure 3b, where a strong resonance can be observed. Furthermore, it is

apparent that the field is restricted to the FP cavity region, which is advantageous for suppressing any cross-sensitivity when using multiplexed sensor elements on a single fiber.

For the practical implementation of such fiber RWGs, it is important to discuss the influence of sensor parameters and possible fabrication errors on the spectral response. Here, the fabrication tolerances are studied within the 2D model, though simplified for higher simulation speed. Nevertheless, these simulations offer useful information for sensor fabrication. First, the total length of the grating was varied, i.e., the two mirrors were set at different grating bars (counted from the center), with the results shown in Figure 4a. It is apparent that a small decrease in mode reflectance is detected for deviations from the optimal positioning of the mirrors. The peak intensity is reduced by ~11.4% if the mirrors are located at the twelfth grating bar compared to the maximum peak intensity that is achieved for the ninth bar. For even longer gratings, progressively more reflected energy leaks into the cladding mode, while, for shorter ones, some of the core mode energy is rejected and excluded from the FP cavity. It is worth noticing that the position of the side peak experiences a blue shift with an increased FP cavity length, while a slight red shift occurred for the main RWG peak, indicating that the overall reflection spectrum behavior can be attributed to some sort of synergistic effect of RWG and FP resonance. For small x-axis mirror positioning errors away from the center of the ninth grating bar, the spectral response of the sensor is shown in Figure 4b. It is apparent that the peak intensities progressively decrease with increased deviations of the mirrors' positions from the optimal ones. For example, a deviation of +100 nm results in a peak reduction to ~59.0% caused by the accumulation of phase errors in FP reflections.

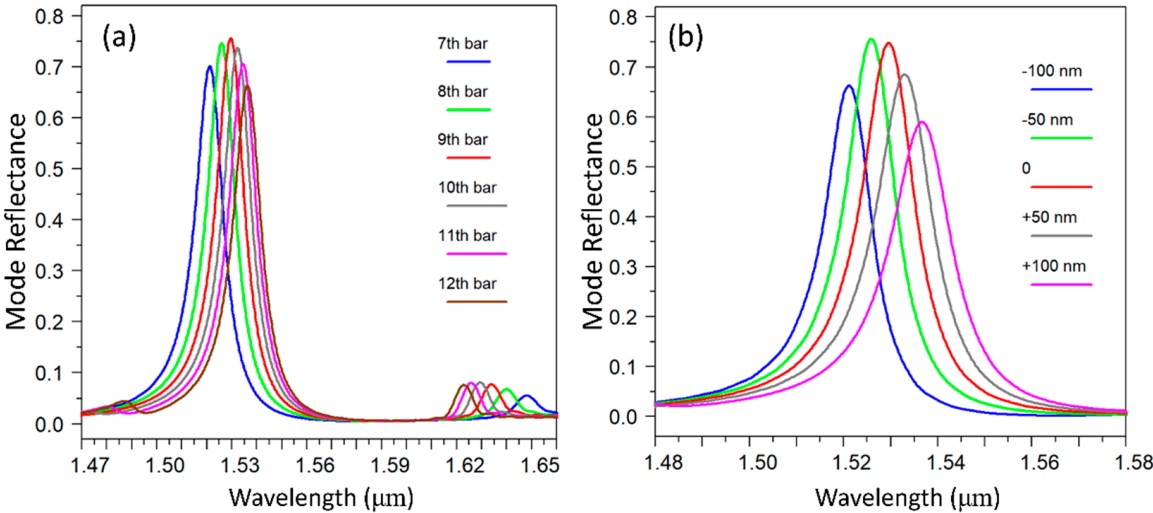

**Figure 4.** Mode reflectance for the designed RWG (**a**) with Au mirrors located at the center of different grating bars, and (**b**) for different deviations of the mirror planes from the center of the ninth grating bar.

Another common error, which may occur in a fabrication process, is the x-axis displacement of the whole RWG structure relative to the center of the fiber core. As shown by the blue curve in Figure 5a, only a small variation in peak intensity occurs with negligible change in peak position for such errors. For example, a reduction in peak intensity of 2.7% is found for an off-center error of 0.8 μm. The reason for this weak dependence is solely the small reduction in coupling efficiency between the reflected beam and the fiber mode. In addition, the influence of the depth h of the trenches used for mirror fabrication (see Figure 1a) was investigated and is shown by a red curve in Figure 5a. The peak intensity remains above 70% as h is reduced to 600 nm and is still above 60%, even if h is reduced to 400 nm, i.e., in the case that the bottom of the trench is already close to the waveguide/fiber interface.

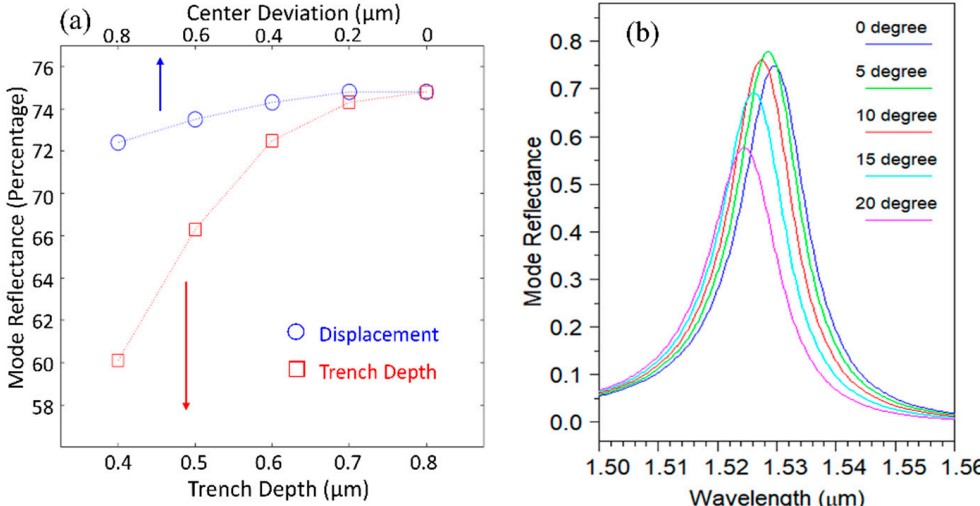

**Figure 5.** (**a**) Mode reflectance for the designed fiber-tip RWG for different off-center x-axis alignment errors (blue, circles) and for different depths of the mirrors' trenches (red, squares); (**b**) Mode reflectance for different inclinations α of the sidewalls of the mirrors' trenches.

As mentioned above in Section 2, the trenches for the Au layer deposition can be fabricated by FIB milling or by etching. For both techniques, a practical problem may arise that the sidewalls are not strictly vertical, but at an angle α relative to the fiber axis. The influence of this sidewall inclination was investigated and is presented in Figure 5b. When introducing tilt angle α, the middle of the mirror plane is fixed. It is observable that the peak intensity is increased by ~4% for α equal to 5°, and slowly decreases with the further increase in α. The reason for the initial increase in peak intensity may be an additional phase accumulation during reflection at the Au layer, which changes the phase matching condition. To summarize, there is a quite tight tolerance of Au mirror positioning with respect to the center of the grating bar, while other fabrication tolerances for the mirror trenches are relatively relaxed.

The performance of a designed practical device was tested by applying 3D FDTD simulations on a high-performance computer server. During simulations, similar x- and z-axis grid sizes were applied in 2D, while y-axis grid was set to 50 nm. The width w of the RWG structure was found to have a negligible influence on the spectral response, and was thus set to be 15 μm. The results for sensor reflectance are presented in Figure 6a. The peak mode reflectance (blue curve) is ~73.3%, which is about 93% that of total reflectance (green curve), showing that, with the assistance of the FP cavity, most of the reflected energy can be collected and coupled into the back-propagating fiber mode. In addition, the peak mode reflectance is at a similar level as that of the 2D result (green dashed curve), with only a slight wavelength shift, showing a good agreement between the 2D and 3D simulation results. The background reflectance is below 3% on both sides of the resonant peak, showing that the designed RWG exhibits a high extinction ratio. The peak mode reflectance is about three times that (~27.1%) for a uniform grating, showing a remarkable enhancement. At the same time, the FWHM is reduced from 23.5 nm to 14.3 nm by employment of the FP cavity.

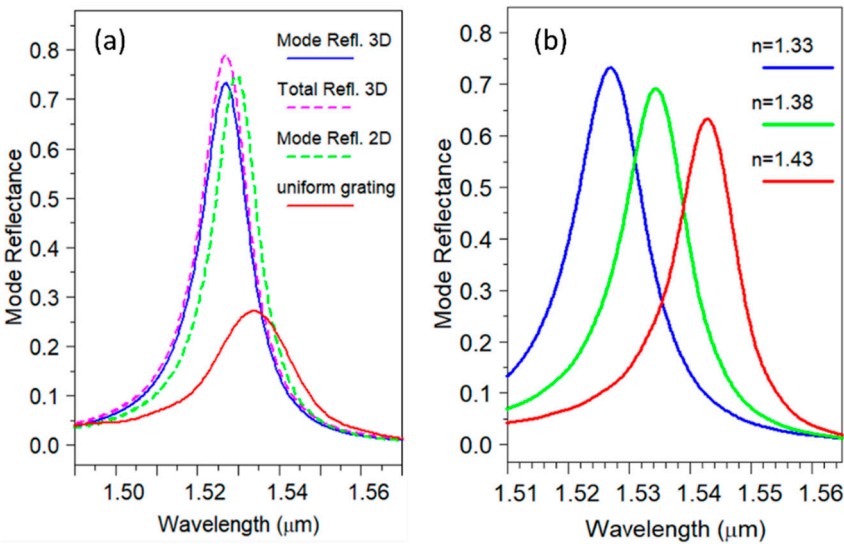

**Figure 6.** (**a**) Mode reflectance (blue), total reflectance (magenta) for a 3D fiber RWG, mode reflectance for a 2D model (green) and mode reflectance of uniform grating (red); (**b**) Spectral response for different ambient refractive indices.

The sensing performance of the designed FP cavity-assisted fiber-tip RWG was investigated by applying 3D FDTD as well. The results are presented in Figure 6b. During simulations, three different ambient refractive indices were used. As the refractive index increases, the resonant peak experiences a linear red shift in the used index range from 1.33 to 1.43. The sensor sensitivity was calculated to be ~158 nm/RIU, like that reported in References [15,20]. The figure of merit (FOM) for resonance type fiber sensors can be defined as the ratio of its sensitivity to FWHM of the resonance peak [27]. It was found that the FOM of the designed RWG sensor was improved by ~64% compared to the uniform grating scenario, mainly due to FWHM narrowing. At the same time, the length of the required grating structure is reduced by approximately one order of magnitude as compared to these recent works, with the grating covering only a small region over the fiber core. With an increase in the ambient medium refractive index, slight reduction in peak intensity is observed, which could be attributed to a reduction of confinement of the guided mode in the RWG layer.

## 4. Conclusions

In this work, an FP cavity-assisted RWG on the tip of a single-mode fiber was studied theoretically for refractive index sensing. In contrast to a uniform RWG directly on the fiber facet, the total grating length can be restricted to a small area around the fiber core while the resonant reflection peak intensity was enhanced by approximately three times, reaching more than 70%. This improved peak intensity is crucial for practical sensing applications as it increases signal-to-noise ratio, thus allowing for a more accurate read-out of peak information. The average sensitivity of the designed fiber-tip RWG sensor is 158 nm/RIU for the refractive index range from 1.33 to 1.43. The improved FOM promises significant application potential in multi-channel biochemical sensing and for real-time medical diagnostics. Such multi-channel sensing can be realized using multi-core fibers, where the compact size of our resonant RWG sensor allows for, e.g., measurements of blood composition changes. In the case of a multiplexed sensor with many RWG sensing elements, the restriction of leaky mode propagation within limited regions around separate cores is important to eliminate cross response between different channels, which is crucial for both spatial and wavelength multiplexing. Furthermore, the new design strategy is preferable for the development of high-performance RWG wavelength filters on a single-mode fiber tip, which may find applications in fiber lasers.

**Author Contributions:** Y.Y.: Conceptualization, Methodology, Writing—Original Draft; Y.X.: Software, Data Curation; N.-K.C.: Supervision; I.P.: Software, Formal analysis; S.S.: Formal analysis, Writing—Review and Editing; D.K.: Supervision, Writing—Review and Editing; Y.R.: Writing—Review and Editing. All authors have read and agreed to the published version of the manuscript.

**Funding:** This work was supported in part by the National Natural Science Foundation of China under Grants 61875247 and 11874243, by Natural Science Foundation of Shandong Province under Grants ZR2020QF086 and ZR2020QA068, by Liaocheng University under Grants 318051411, 319190301 and 31805180101, and by Deutsche Forschungsgemeinschaft (DFG Ki482/19-1).

**Acknowledgments:** Yicun Yao acknowledges the support of Introduction and Cultivation Plan of Youth Innovation Talents for Universities of Shandong Province.

**Conflicts of Interest:** The authors declare no conflict of interest.

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
