# Peer review of "Design of Fiber-Tip Refractive Index Sensor Based on Resonant Waveguide Grating with Enhanced Peak Intensity"

_applsci, doi:10.3390/app11156737_

Round 1

Reviewer 1 Report

The paper presents a theoretical fiber tip sensor design.  The design has a complex structure including proposed use of "lithographic" steps.  I'm not sure how you do lithography on the end of a fiber.  The paper would benefit from a process flow diagram showing clearly how steps are proposed to be performed.  I will accept the paper when this figure is added, if the steps shown are believable.

Reviewer 2 Report

The authors report on the design of a refractive index sensor using a resonant waveguide grating on a fiber tip. I understand the work has merit and the results are adequately shown. In the following, I list a set of suggestions that may help the authors to improve the quality of their manuscript.

  1. It seems that the first paragraph in section 2 (Structure and working principle) remained from the paper template. Please remove it.
  2. The authors say that the side peak in Fig. 3a can be attributed to the “FP resonance behavior”. I am afraid this statement is somehow vague. Please develop on it. Is there any chance that this side peak comes from higher-order diffraction in the grating?
  3. Axis description in Fig. 5 a is missing.
  4. The authors inform that the sensor sensitivity is similar to the ones found in references [15-20]. I suggest that the authors also analyze the (possible) improvement on the system’s resolution. By doing this, they can evaluate the effect of reducing the FWHM of the resonance peak (with respect to a uniform grating scenario). It would also be interesting if the authors could include an assessment of the Sensitivity/FWHM ratio in their analyses.

Round 2

Reviewer 1 Report

okay, I think they can make it

Author Response

We appreciate for the positive comment from the reviewer. We have polished the English carefully, with grammatical and formatting errors corrected. All the changes are marked (by the “Track Changes” function of MS Word) in the revised manuscript.